# Cell type-specific pharmacology of NMDA receptors using masked MK801

Yunlei Yang[†], Peter Lee, Scott M Sternson*

Janelia Research Campus, Howard Hughes Medical Institute, Ashburn, United States

**Abstract** N-Methyl-D-aspartate receptors (NMDA-Rs) are ion channels that are important for synaptic plasticity, which is involved in learning and drug addiction. We show enzymatic targeting of an NMDA-R antagonist, MK801, to a molecularly defined neuronal population with the cell-type-selectivity of genetic methods and the temporal control of pharmacology. We find that NMDA-Rs on dopamine neurons are necessary for cocaine-induced synaptic potentiation, demonstrating that cell type-specific pharmacology can be used to dissect signaling pathways within complex brain circuits.

## Introduction

N-Methyl-D-aspartate receptors (NMDA-Rs) are glutamate-gated ion channels that are critical for the regulation of synaptic functions in the central nervous system, such as synaptic plasticity (*Malenka and Nicoll, 1993*; *Collingridge et al., 2004*). NMDA-R dependent synaptic plasticity plays an important role in learning. This includes learning that can also have maladaptive consequences, for example sensitization of drug-related behaviors (*Kalivas and Alesdatter, 1993*; *Ungless et al., 2001*). However, because NMDA-Rs are expressed in most cell types in the brain (*Conti et al., 1997*; *Verkhratsky and Kirchhoff, 2007*), it is a considerable challenge to selectively assess the importance of NMDA-R mediated synaptic plasticity in specific cell types.

The functional contribution of NMDA-Rs to physiology and behavior can be examined using either genetic or pharmacological methods. Small molecule antagonists allow rapid blockade of NMDA-Rs, which has been critical for analyzing synaptic plasticity processes that operate over short timescales. However, existing pharmacological agents are not cell type-selective. Alternatively, genetic methods can be used, typically involving Cre-recombinase (Cre)-mediated excision of a loxP flanked exon in *Grin1*, which encodes a critical NMDA-R subunit that is required for channel function. This allows cell type-specific ablation of NMDA-R function by crossing to a mouse line with cell type-selective expression of Cre or by spatially targeted injection of Cre-expressing viral vectors. A major limitation is that genetic methods are not fast; they typically disrupt NMDA-Rs over timescales ranging from a week to the lifetime of an animal, during which time substantial compensatory effects in circuit function are observed (*Engblom et al., 2008*; *Zweifel et al., 2008*). Here, we describe an approach to combine the cell type-selectivity of genetic methods with the temporal control of pharmacology by targeting a small molecule NMDA-R antagonist to molecularly defined neuronal cell types (*Figure 1A*).

## Results

### Masked MK-801

MK801 is a potent non-competitive antagonist of NMDA-Rs that use-dependently blocks the channel in the open (glutamate-bound) state (*Wong et al., 1986*; *Woodruff et al., 1987*). Furthermore, MK801 can bind and block the NMDA-R channel ion conductance pore from the intracellular compartment (*Berretta and Jones, 1996*; *Bender et al., 2006*). To develop a cell type-specific pharmacological strategy to rapidly block NMDA-Rs in subpopulations of neurons within brain tissue,

*For correspondence:
sternsons@janelia.hhmi.org

Present address: [†]Department of Neuroscience and Physiology, State University of New York Upstate Medical University, Syracuse, United States

Competing interests: The authors declare that no competing interests exist.

**eLife digest** Learning is critical to survival for humans and other animals. The learning process is regulated by receptors on the surface of brain cells called N-Methyl-D-aspartate receptors (or NMDA receptors for short). These receptors help to strengthen signals between brain cells, which allows a new concept or action to be learned. However, it has been difficult to pin down how the role of NMDA receptors selectively in specific types of brain cells. While drugs can be used to quickly block NMDA receptors throughout the brain, it is hard to target drugs to a specific cell type. Also, genetic engineering can be used to selectively knock out NMDA receptors in certain types of brain cells, but these techniques are too slow, and can take weeks or even a lifetime to work.

Now, Yang et al. have developed a clever way to combine an NMDA-blocking drug and genetic engineering to study NMDA receptors' responses to cocaine in specific brain cells. This approach involved first creating an inactive form of an NMDA-blocking drug that can only becomes active when it is processed by an enzyme that is normally produced in pigs' livers. Next, living mouse brain cells, including some that were engineered to express the pig enzyme, were exposed to the drug in the laboratory. The drug blocked the NMDA receptors on brain cells that expressed the enzyme, but not the receptors on nearby brain cells that lacked the enzyme. This occurred even though all the cells produced NMDA receptors and all were exposed to the drug.

NMDA receptors have been known to play an important role in cocaine addiction for more than 20 years. Drugs like cocaine can co-opt the normally healthy learning process involving NMDA receptors and lead to a maladaptive form of learning that is commonly called addiction. Cocaine strengthens signals between brain cells causing the behaviors associated with using cocaine to become deeply ingrained and difficult to change. Yang et al. used cell type-specific targeting of a drug that blocks NMDA receptors to observe what happened in cocaine-exposed brain cells with, or without, working NMDA receptors.

As expected, the experiments showed that cocaine didn't strengthen brain signals in cells without working NMDA receptors. Specifically, the experiments showed that NMDA receptors on a type of brain cell that release a pleasure-inducing chemical called dopamine are necessary for cocaine–induced synaptic plasticity. The combination technique developed by Yang et al. will likely be used by other scientists to further study the role of NMDA receptors in specific brain cells during addiction and normal brain activity.

we synthesized an inert masked MK801 derivative that could be unmasked inside cells by transgenically expressed porcine liver esterase (PLE) (*Figure 1B*) (*Tian et al., 2012*) . A carboxymethylpropyl ester group has been used previously in brain tissue to mask a fluorophore and a kinesin inhibitor, and extensive analysis showed that this group was stable to endogenous neuronal esterases but was cleaved in neurons transgenically expressing PLE (*Tian et al., 2012*). Because MK801 contains a secondary amine that is critical for NMDA-R antagonism, we used a 4-hydroxy-3-nitrobenzyl carbamate (*Gesson et al., 1994*) to link MK801 and a carboxymethylpropyl ester to generate CM-MK801 (*Figure 1C*). Enzymatic cleavage of the CM ester inside neurons expressing PLE leads to spontaneous 1,6-elimination of the vinylogous hemiaminal, liberating MK801 (*Figure 1D*).

## Cell type-specific NMDA-R blockade

Three critical properties of the CM-MK801/PLE system must be evaluated for cell type-specific pharmacology: 1) CM-MK801 should be efficacious and completely block NMDA-Rs in PLE$^+$ neurons, 2) PLE$^-$ neurons should not unmask CM-MK801, and 3) unmasked MK801 should be confined to PLE$^+$ neurons without affecting adjacent PLE$^-$ neurons. To examine the cell type selectivity of this reagent, we measured NMDA-R excitatory postsynaptic currents (EPSCs) in brain slices from the mouse cerebral cortex. In utero electroporation (IUE) of the bicistronic plasmid *pCAG::PLE-IRES-mCherry* into the developing mouse brain allowed PLE and mCherry co-expression in only a subset of cortical layer 2/3 pyramidal neurons. Consistent with previous work, PLE expression was well tolerated in cortical neurons (*Tian et al., 2012*). We isolated NMDA-R EPSC responses for cortical layer 4→layer 2/3 (L4→L2/3) projections by recording neurons in L2/3 while electrically stimulating presynaptic neurons in L4 during constitutive pharmacological block of other ionotropic glutamate receptors (AMPA and

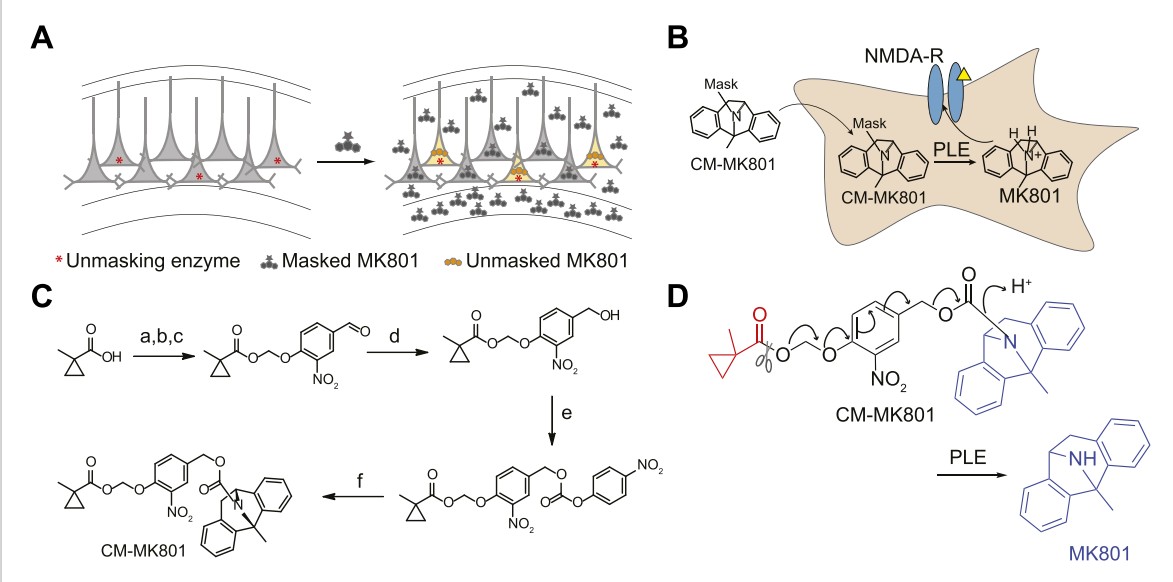

**Figure 1**. Cell type specific pharmacology for NMDA-R inhibition. (**A**, **B**) Strategy for cell type-specific targeting of a masked MK801 molecule to a defined subpopulation of neurons in brain tissue that transgenically express an unmasking enzyme. The masked MK801 enters every cell but MK801 is unmasked in only those cells that transgenically express the enzyme porcine liver esterase (PLE). (**B**) The intracellularly liberated MK801 can block NMDA-Rs in the PLE-expressing neurons. Yellow triangle, glutamate. (**C**) Synthesis of CM-MK801. (a) chloromethylchlorosulfate; (b) NaI; (c) 4-hydroxy-3-nitrobenzaldehyde, 44% for a-c; (d) NaBH$_4$, 48%; (e) 4-nitrophenyl chloroformate, 65%; (f) MK801, 32%. (**D**) Enzymatic hydrolysis of CM-MK801 by PLE is followed by spontaneous 1,6-elimination to liberate MK801.

kainate receptors) and GABA receptors. Brain slices were incubated with CM-MK801 (5 μM), and L4→L2/3 NMDA-R EPSCs were recorded using the perforated patch configuration (necessary to avoid dilution of enzyme-liberated MK801 by diffusion into the patch pipette that was observed with whole cell recordings) in mCherry-expressing (PLE$^+$) and adjacent non-expressing (PLE$^-$) neurons (*Figure 2A,B*). The NMDA-R EPSC amplitude (*Figure 2C*) and mean NMDA-R charge transfer (Q$_{NMDA}$: mean integrated current, *Figure 2E*) decreased in a use-dependent manner in PLE$^+$ neurons to a level similar to treatment with the competitive NMDA-R antagonist D-(−)-2-amino-5-phosphonopentanoic acid (AP5, 100 μM) (*Figure 2E–G*; Q$_{NMDA}$ inhibition: CM-MK801, 82% ± 4; AP5, 84% ± 4; n = 11; unpaired t test, p = 0.75). This shows that CM-MK801 can completely block NMDA-R currents in neurons expressing the PLE transgene. In contrast, in the presence of CM-MK801, PLE$^-$ neurons that were adjacent to PLE$^+$ neurons showed scant reduction of Q$_{NMDA}$ (*Figure 2D,F*). The slight Q$_{NMDA}$ reduction in PLE$^-$ neurons exposed to CM-MK801 was not significantly different than Q$_{NMDA}$ in neurons in the absence of CM-MK801 (Q$_{NMDA}$ inhibition: PLE$^-$/CM-MK801: 16% ± 8.9, n = 6; PLE$^+$: 33% ± 8.6, n = 4; PLE$^-$: 21% ± 6.3; ANOVA, $F_{2,13}$ = 2.3, p = 0.15), thus Q$_{NMDA}$ reduction in PLE$^-$ neurons was due to modest NMDA-R synaptic rundown (*Rosenmund and Westbrook, 1993*) and not an effect of MK801 in PLE$^-$ neurons. These results show that CM-MK801 can be targeted specifically to neurons expressing a transgenic esterase and that CM-MK801 is selective, where MK801, once liberated, is confined to the PLE-expressing cell population.

## Cell type-specific NMDA-R pharmacology in cocaine-induced synaptic plasticity

An application of cell type-specific pharmacology for NMDA-R is to examine synaptic plasticity in molecularly defined neurons. One problem that has been difficult to address with existing tools is the role of NMDA-Rs in cocaine-induced synaptic plasticity. Cocaine blocks dopamine reuptake, which elevates extracellular concentrations. Earlier work demonstrated that cocaine and elevated dopamine leads to long-term synaptic potentiation of AMPA receptor-type (AMPA-R) synaptic currents in dopamine (DA) neurons in the ventral tegmental area (VTA) (*Ungless et al., 2001*; *Argilli et al., 2008*)

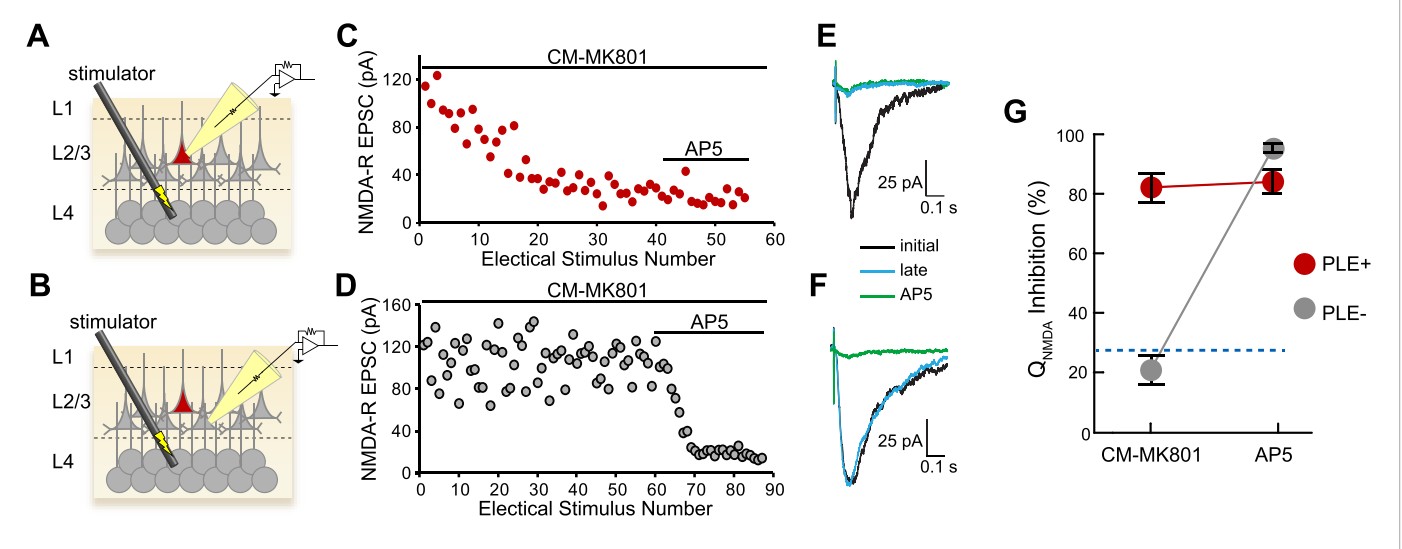

**Figure 2**. Cellular selectivity of CM-MK801/PLE ester/esterase pair. (**A**, **B**) Schematic diagrams of the experimental procedure. Perforated patch voltage clamp recordings of NMDA-R synaptic currents were made on both (**A**) PLE/mCherry-expressing (PLE+, red) layer 2/3 (L2/3) cortical neurons and (**B**) adjacent neurons lacking PLE (PLE−, gray) while electrically stimulating presynaptic neurons in layer 4 (L4) in the presence of CM-MK801. (**C**, **D**) Treatment of the brain slices with CM-MK801 (5 µM) during L4→L2/3 synaptic stimulation showed gradual use-dependent reduction of the NMDA-R EPSC amplitude in (**C**) PLE+ but not in (**D**) PLE− neurons, indicating that the CM-MK801 was converted to MK801 selectively in PLE+ neurons without spilling over to adjacent PLE− neurons. Subsequent addition of the competitive NMDA-R antagonist, AP5, suppressed the NMDA-R EPSC in PLE− neurons but not in PLE+ neurons, which were fully blocked by CM-MK801. Electrical stimuli were delivered every 15 s. (**E**, **F**) Overlaid NMDA-R averaged excitatory postsynaptic currents (EPSCs) from (**E**) PLE+ and (**F**) PLE− neurons showing the initial response (black), responses after electrical stimulation (late, blue) and the response in the presence of AP5 (green). Electrical stimulation artifact reduced for clarity. (**G**) Grouped data for NMDA-R EPSC charge transfer ($Q_{NMDA}$) inhibition for PLE+ (n = 14) and PLE− (n = 17) neurons during CM-MK801 treatment and subsequent exposure to AP5. For comparison, dotted line shows $Q_{NMDA}$ inhibition in PLE+ and PLE− neurons in the absence of CM-MK801, which is due to modest synaptic rundown. Data is represented as mean and error bars indicate s.e.m.

(*Figure 3A*), a key neuronal cell type associated with reward and addiction. Further analysis demonstrated that synaptic potentiation involved signaling through dopamine receptor 5, new protein synthesis, and insertion of AMPA-R subunits (*Argilli et al., 2008*). These experiments also showed that NMDA-R blockade, by systemic application of MK801, prevented synaptic potentiation in DA neurons (*Ungless et al., 2001*) (*Figure 3A*). However, it has been challenging to establish the functional importance of NMDA-Rs specifically in DA neurons for cocaine-induced synaptic AMPA-R potentiation, as opposed to a possible indirect process involving NMDA-Rs on other cell types. To disrupt this process specifically in DA neurons, studies using dopamine neuron-specific knockout of *Grin1* were performed but, in several instances, these were confounded by compensatory effects in which synaptic AMPA-Rs were constitutively potentiated in DA neurons in the absence of cocaine (*Engblom et al., 2008*; *Zweifel et al., 2008*). Even virally mediated deletion of *Grin1* showed these compensatory effects within 1 week (*Zweifel et al., 2008*). In contrast, another study found that use of tamoxifen-activated Cre-ER for selective *Grin1* ablation in DA neurons prevented a cocaine-induced shift in the rectification of AMPA-Rs after 1 week (*Engblom et al., 2008*), which indicated that dopamine neuron NMDA-Rs were required for cocaine-induced plasticity.

We sought to investigate cocaine-induced synaptic plasticity by rapid blockade of NMDA-Rs with the temporal control of pharmacology and the cell-type selectivity of genetics by directing MK801 to DA neurons using cell type-specific pharmacology. For this, we co-expressed PLE and the fluorescent protein mCherry in DA neurons using a Cre-dependent recombinant adeno-associated viral vector, *rAAV2/1-Synapsin::FLEX-rev-PLE-2a-mCherry* (2a: ribosomal skip sequence [*Donnelly et al., 2001*]) that was targeted unilaterally into the VTA of *Slc6a3^Cre* mice, a mouse line that selectively expresses Cre-recombinase in DA neurons (*Zhuang et al., 2005*). In the same mice, control (PLE−) DA neurons were labeled using a Cre-dependent virus expressing EGFP that was targeted to DA neurons in the

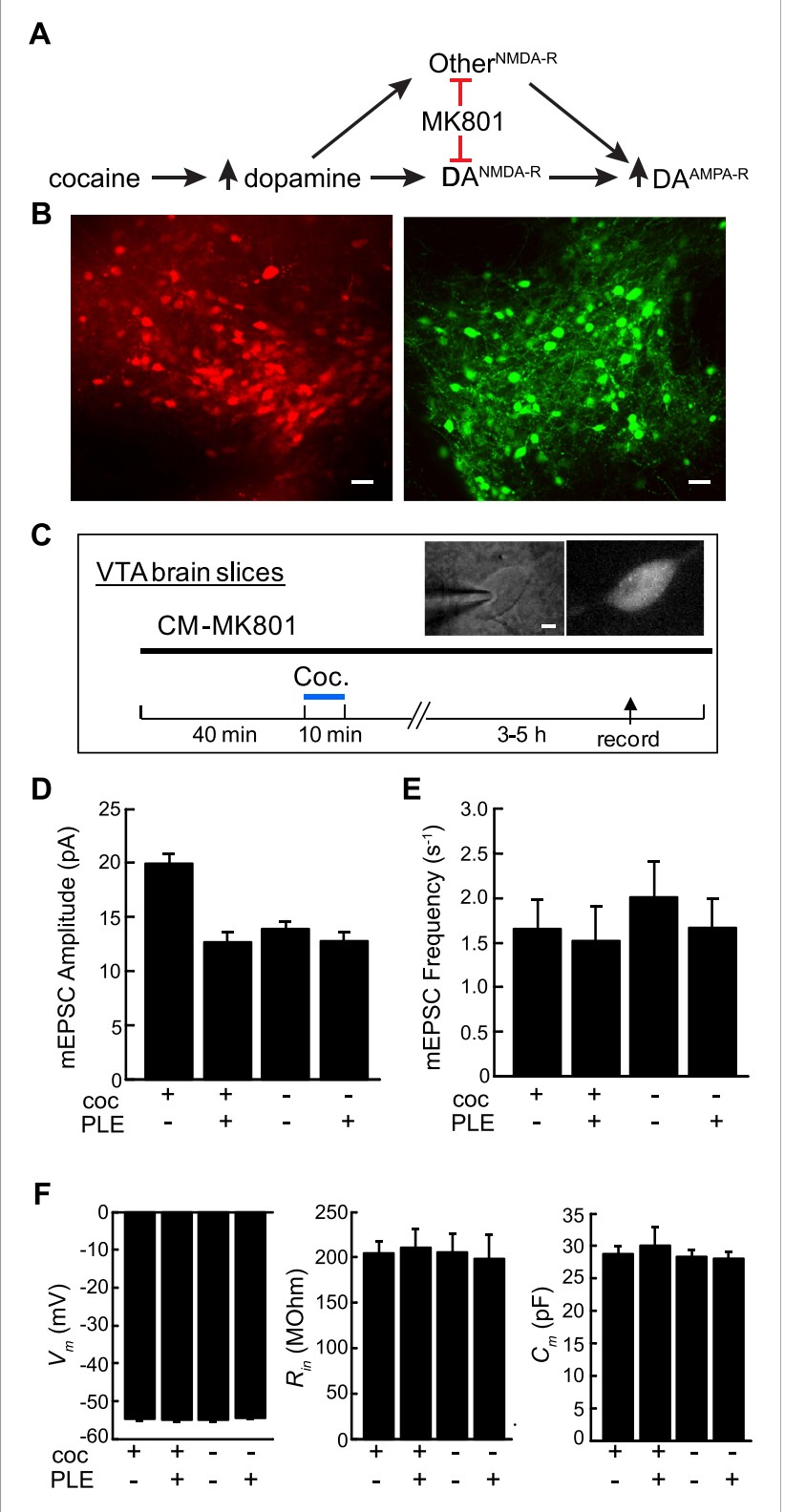

**Figure 3**. Cell type-selective blockade of cocaine-induced synaptic plasticity in dopamine neurons. (**A**) Schematic diagram of cocaine-induced synaptic plasticity. Cocaine increases extracellular dopamine which leads to NMDA-R dependent upregulation of synaptic AMPA (α-amino-3-hydroxy-5-methyl-4-isoxazolepropionic acid) receptors. It is unclear if this is through NMDA receptors on DA neurons or through NMDA-R on other cell types that may have a

*Figure 3. continued on next page*

*Figure 3. Continued*

subsequent effect on AMPA-R activity in DA neurons. (**B**) Confocal images of ventral tegmental area (VTA) brain slices from dopamine transporter Cre (*Slc6a3^{Cre}*) mice transduced with Cre-dependent viruses expressing either PLE and mCherry (left) or EGFP (right) on opposite sides of the VTA. Scale, 50 μm. (**C**) Experimental protocol where VTA-containing brain slices were incubated with CM-MK801 for 40 min, followed by addition of cocaine for 10 min and washout. 3–5 hr later, miniature excitatory postsynaptic AMPA-R currents (mEPSCs) in DA neurons expressing either GFP (PLE⁻) or mCherry (PLE⁺) were recorded. Inset, images of (left) patch pipette recording from dopamine neuron (right) expressing mCherry and PLE. Scale, 5 μm. (**D**) For brain slices treated with CM-MK801, AMPAR-mediated mEPSC amplitude was increased in PLE⁻ VTA DA neurons following cocaine treatment. Synaptic potentiation was blocked in PLE⁺ neurons treated with cocaine, and the mEPSC amplitude was similar to that in PLE⁺ or PLE⁻ neurons that were not exposed to cocaine. (**E**) mEPSC frequency was unaffected by cocaine or CM-MK801. (**F**) Resting neuronal membrane potential ($V_m$), input resistance ($R_{in}$), and membrane capacitance ($C_m$) were not changed by expression of PLE, or exposure to cocaine and CM-MK801. Data is represented as mean and error bars indicate s.e.m.

VTA on the other side of the brain. To investigate the involvement of dopamine neuron NMDA receptors, we adopted a previously described brain slice model that was sufficient to recapitulate the processes underlying cocaine-mediated synaptic AMPA-R potentiation in DA neurons ex vivo (*Argilli et al., 2008*). VTA brain slices containing DA neurons expressing PLE/mCherry and EGFP on opposite sides of the brain (*Figure 3B*) were incubated with CM-MK801 (5 μM) in artificial cerebrospinal fluid (ACSF) followed by a short exposure to cocaine (5 μM in ACSF, 10 min) or vehicle and afterwards the brain slices were transferred to ACSF with CM-MK801 for 3–5 more hours. After this induction period, miniature excitatory postsynaptic AMPA-R currents (mEPSCs) were recorded from DA neurons, which could be isolated by blocking inhibitory GABA-Rs and voltage gated sodium channels. As shown previously in VTA-containing brain slices exposed to cocaine (*Argilli et al., 2008*), DA neurons showed significantly elevated potentiation of mEPSC amplitudes in PLE⁻/GFP⁺ DA neurons in the presence of CM-MK801, which further demonstrates the lack of activity for CM-MK801 in PLE⁻ neurons. Strikingly though, for PLE⁺/mCherry⁺ DA neurons in CM-MK801 and cocaine, mEPSC amplitudes were not elevated and were similar to PLE⁺ or PLE⁻ DA neurons in CM-MK801 that had not been exposed to cocaine. (*Figure 3D*; Coc⁺/PLE⁻, 19.9 ± 0.9 pA, n = 17; Coc⁺/PLE⁺, 12.7 ± 0.8 pA, n = 14; Coc⁻/PLE⁻, 13.9 ± 0.6 pA, n = 12; Coc⁻/PLE⁺, 12.8 ± 0.7 pA, n = 15; ANOVA $F_{3,57}$ = 19.0, p < 0.001). The frequency of mEPSCs in DA neurons was not affected by CM-MK801 or cocaine (*Figure 3E*, ANOVA $F_{3,57}$ = 0.30, p = 0.82). In addition, the membrane properties of DA neurons after CM-MK801 exposure were indistinguishable in PLE⁺ or PLE⁻ cells treated or untreated with cocaine, further indicating that CM-MK801 is not perturbative to resting neuronal properties. Therefore, using cell type-specific pharmacology in an ex vivo preparation, these experiments demonstrate that NMDA-Rs in DA neurons are necessary for cocaine-induced potentiation of synaptic AMPA-R currents in this cell type ex vivo, which is consistent with a prior report that examined this problem using Cre-ER in vivo (*Engblom et al., 2008*).

## Discussion

Taken together, these experiments demonstrate the effectiveness of a cell type-specific pharmacology approach to selectively block NMDA-Rs in a molecularly defined neuron population using an esterase–ester pair. Cell type-specific pharmacology for activating genetically engineered ion channels has been described (*Slimko et al., 2002*; *Magnus et al., 2011*), and here we demonstrate rapid cell type-specific blockade of a native ion channel with a small molecule. Our data indicates that MK801 was liberated only in PLE⁺ but not in neighboring PLE⁻ neurons, showing that this approach can be used to block NMDARs with cellular specificity. The absence of a neighbor effect with CM-MK801/PLE is consistent with experiments targeting MK801 to individual neurons in a patch pipette (*Bender et al., 2006*). We applied this to DA neurons to further examine the role for NMDA-Rs in cocaine-induced synaptic potentiation, which has been investigated in several pharmacological (*Ungless et al., 2001*; *Argilli et al., 2008*) and genetic studies (*Engblom et al., 2008*; *Zweifel et al., 2008*). Our experiments using cell type-specific pharmacology provide additional evidence for the necessity of NMDA-Rs in cocaine-induced long-term synaptic potentiation in DA neurons ex vivo. One limitation of the approach presented here is that the aqueous solubility of CM-MK801 (5–10 μM) make

it less suitable for direct injection into the brain because of the requirement for elevated levels of co-solvents, such as dimethyl sulfoxide (DMSO) (>0.1%), which resulted in VTA-containing brain slices that were unsuitable for recordings. Because of this, it was not possible for us to determine if AMPA-R potentiation occurs after inactivation of NMDA-R in vivo using the chemogenetic method described here. Therefore, additional improvements to cell type-specific pharmacological techniques are needed to facilitate manipulations in mammalian brains in vivo.

However, because functional NMDA receptors are widely expressed throughout the mammalian nervous system, ex vivo approaches to rapidly disrupt NMDA-R signaling with cell type specificity have broad utility. NMDA-Rs are expressed in both neurons and glia (*Conti et al., 1997*; *Verkhratsky and Kirchhoff, 2007*), and selective genetic access to these cell classes in the brain could allow for dissection of their relative role in synaptic function. Moreover, neurons have been classified into many different molecularly defined cell types, which are accessible through a growing resource of Cre-expressing mouse lines (*Gong et al., 2007*). In combination with Cre-dependent viral approaches (*Atasoy et al., 2008*), this will allow the role of NMDA-Rs to be selectively evaluated in any molecularly defined neuron population in manner similar to what we demonstrated for DA neurons. In addition, the relative roles of presynaptic and postsynaptic NMDA-Rs has been extensively analyzed in brain slices (*Corlew et al., 2008*; *Rodriguez-Moreno and Paulsen, 2008*), which PLE/CM-MK801 may facilitate because it offers a new method for selectively blocking NMDA-Rs in defined subpopulations of neurons. Finally, the strategy applied here for selective delivery of MK801 can be extended to other small molecules for use in brain tissue (*Tian et al., 2012*), allowing for an approach that combines chemistry, molecular genetics, and neurobiology to dissect cell signaling pathways in specific cell types in the central nervous system.

## Materials and methods

All experimental protocols were conducted according to United States National Institutes of Health guidelines for animal research and were approved by the Institutional Animal Care and Use Committee at Janelia Research Campus. Animals were housed on a 12 hr light (06:00)/dark (18:00) cycle with ad libitum access to water and mouse chow (PicoLab Rodent Diet 20, 5053 tablet, TestDiet, St. Louis, MO, United States). Experiments with cocaine-mediated synaptic plasticity (*Figure 3*) were done with the experimenter blind as to whether a cocaine solution or vehicle was being applied to the brain slice.

### IUE procedures

As previously described (*Saito and Nakatsuji, 2001*), timed pregnant female C57Bl/6NCrl mice were purchased from Charles River laboratories and E15 embryos were electroporated following pressure microinjection (Picrospritzer) into the right lateral ventricle of a DNA mixture (0.5 µg/µl, approximately 200 nl; *pCAG::PLE-IRES-mCherry*) encoding PLE and a mCherry signal marker. Postoperative analgesia was provided (buprenorphine was administrated intraperitoneally at a dose of 0.1 mg/kg along with ketoprofen administrated subcutaneously at a dose of 5 mg/kg).

### Electrophysiology of layer 2/3 neurons in cortical brain slices

Electroporated mice were sacrificed between postnatal days 14–21 for brain slice recordings from transfected layer 2/3 cortical pyramidal neurons. As previously described (*Yang et al., 2011*), animals were deeply anesthetized with isofluorane, decapitated and the brains removed in ice cold sectioning solution. The brains were then mounted on a stage using Krazy Glue, and coronal brain slices (300 µm thickness) were cut using a cooled tissue slicer (Vibratome, 2000). Slices were prepared in chilled cutting solution containing the following (in mM): 110 choline chloride, 2.5 KCl, 1.25 mM $NaH_2PO_4$, 2 $CaCl_2$, 7 $MgSO_4$, 25 D-glucose, 3.1 Na-pyruvate, and 11.6 Na-L-ascorbate, aerated with 95% $O_2$ / 5% $CO_2$. Brain slices were transferred to the chamber containing (in mM): 119 NaCl, 25 $NaHCO_3$, 11 D-glucose, 2.5 KCl, 1.25 $MgCl_2$, 2 $CaCl_2$, and 1.25 $NaH_2PO_4$, aerated with 95% $O_2$/5% $CO_2$ at 34 °C for 30 min and moved to room temperature until transferred to a recording chamber on the stage of an Olympus (Tokyo, Japan) BW51XI microscope.

Electrophysiological recordings were performed as described previously (*Yang et al., 2011*) using a multiple 700B amplifier (Molecular Devices). Slices were transferred to the recording chamber on the stage of an Olympus BW51XI microscope and perfused at a rate of 1–2 ml/min with ACSF (28–30°C)

containing the following (in mM): 119 NaCl, 25 NaHCO$_3$, 11 D-glucose, 2.5 KCl, 1.25 MgCl$_2$, 2 CaCl$_2$, and 1.25 NaH$_2$PO$_4$, aerated with 95% O$_2$/5% CO$_2$, also 5 μM MK-801. Perforated patch recordings were made on either PLE/mCherry expressing neurons expressing neurons or adjacent non-expressing neurons using electrodes with tip resistances of 4–5 MΩ immediately after circulating the ACSF with CM-MK801. 40 to 50 min later, ACSF was switched to 0 Mg$^{2+}$ in the presence of picrotoxin (50 μM) and CNQX (10 μM). The intracellular solution for voltage-camp recordings contained the following (in mM): 125 CsCl, 5 NaCl, 10 HEPES, 0.6 EGTA, 4 Mg-ATP, 0.3 Na$_2$GTP, 10 lidocaine N-ethyl bromide (QX-314) and Gramicidin, pH 7.35 and adjusted to 290 mOsm. Gramicidin (Sigma, St. Louis, MO, United States) was dissolved in dimethylsulfoxide (Sigma) (1–2 mg/ml) then diluted in the pipette filling solution to a final concentration of 0.2–5 μg/ml. The holding potential for voltage-clamp recordings was—70 mV, and responses were digitized at 10 kHz. NMDAR-EPSCs were elicited by 0.1 ms electrical stimuli (100–300 μA) every 15 s with an isolated pulse stimulator (WPI) using a concentric bipolar electrode (FHC, CBAEC75) placed at layer 4 of cortex. 2-amino-5-phosphonovaleric acid (100 μM, AP5) was added in the end of most experiments.

## Electrophysiology of VTA DA neurons

Experimental techniques were similar to those reported previously (*Argilli et al., 2008*). 9–10 days after viral injection, mice expressing Cre recombinase in DA neurons (*Slc6a3$^{Cre}$* mice) were deeply anaesthetized with isofluorane and decapitated. Coronal brain slices (200 μm) containing VTA were prepared in chilled cutting solution containing (in mM): 110 choline chloride, 2.5 KCl, 1.25 mM NaH$_2$PO$_4$ , 2 CaCl$_2$, 7 MgSO$_4$, 25 D-glucose, 3.1 Na-pyruvate, and 11.6 Na-L-ascorbate, aerated with 95% O$_2$ / 5% CO$_2$. Brain slices were incubated at 34 °C for 30 min in ACSF containing (in mM): 119 NaCl, 25 NaHCO$_3$, 11 D-glucose, 2.5 KCl, 1.25 MgCl$_2$, 2 CaCl$_2$, and 1.25 NaH$_2$PO$_4$, aerated with 95% O$_2$/5% CO$_2$ and then transferred to the chamber at room temperature containing ACSF with CM-MK801 (5 μM) for 40 min. Cocaine (5 μM) dissolved in ACSF or ACSF alone were added to the chamber. After 10 min, the brain slices were rinsed twice and transferred to another chamber containing ACSF and CM-MK801. Electrophysiological recordings were performed 3–5 hr later. The brain slices were transferred to a recording chamber on the stage of an Olympus (Tokyo, Japan) BW51XI microscope. Dopamine neurons were identified by fluorescence of either mCherry (*via* rAAV-*Synapsin*::FLEX-*rev*-PLE-2a-mCherry virus) or EGFP (*via* rAAV-*CAG*::FLEX-*rev*-EGFP virus) and hyperpolarization-induced currents (*I*$_h$), and then visually targeted with infrared gradient contrast optics Whole-cell recordings were made on DA neurons using a Multiclamp 700B amplifer (Molecular Devices, Sunnyvale, CA, United States). Neurons were voltage-clamped at −70 mV. mEPSCs were recorded from DA neurons in the presence of picrotoxin (50 μM) and tetrodotoxin (TTX, 1 μM). Responses were digitized at 10 kHz. Series resistances were less than 30 MOhm. The intracellular solution for voltage-clamp recordings contained the following (in mM): 125 CsCl, 5 NaCl, 10 HEPES, 0.6 EGTA, 4 Mg-ATP, 0.3 Na$_2$GTP, and 10 lidocaine N-ethyl bromide (QX-314), pH 7.35 and 290 mOsm.

## Data analysis

For detection of mEPSCs, we used template matching (Clampfit, Molecular Devices) followed by visual inspection. Statistical tests were performed in Microsoft Excel or SigmaPlot. Data is represented as mean ± s.e.m., and error bars indicate s.e.m.

## Viral injections

Mice expressing Cre recombinase in DA neurons (*Slc6a3$^{Cre}$* mice) (P21-P24) were anaesthetized with isoflurane and placed into a stereotaxic apparatus (David Kopf Instruments, Tujunga, CA, United States). The skull was exposed via a small incision and two small holes were drilled on either side of the midline for viral injection. A pulled-glass pipette with 20–40 μm tip diameter was inserted into the brain and two injections (60 nl) of the rAAV2/1-*Synapsin*::FLEX-*rev*-PLE-2a-mCherry or rAAV-*CAG*::FLEX-*rev*-EGFP virus were made at coordinates around the VTA (coordinates, bregma: −2.5 mm; midline: ±0.5 mm; skull surface: −4.5 mm and −5.0 mm). Mice were returned to their home cage typically for 9 or 10 days to recover and for expression of PLE and mCherry or EGFP. A micromanipulator (Narishige) was used to control injection speed at 30 nl/min, and the pipette was withdrawn 15 min after the final injection.

## Chemical synthesis

**1**

Preparation of chloromethyl 1-methylcyclopropanecarboxylate (1); 1-methyl cyclopropane-1-carboxylic acid (4.0 g, 40.0 mmol) was added to mixture of potassium carbonate (22.1 g, 160 mmol), tetra butyl ammonium hydrogen sulfate (1.36 g, 4.0 mmol), water (50 ml), and dichloromethane (DCM) (40 ml) at room temperature. After stirred for 10 min, solution of chloromethyl chlorosulfate in DCM (40 ml) was slowly added and stirred for 5 hr. The reaction mixture was diluted with 150 ml of water and extracted with DCM (3 × 100 ml), combined organic layer was washed with saturated brine (100 ml), dried over $MgSO_4$, filtered, and concentrated to an oil. The oil was filtered through a pad of silica gel with 150 ml of DCM and concentrated to give crude chloromethyl 1-methylcyclopropanecarboxylate (3.8 g, 64%). $^1$H NMR, $\delta_H$ (CDCl$_3$, 400 MHz) 5.70 (2H, s), 1.33 (3H, s), 1.33 (2H, dd), 0.79 (2H, dd, J = 5.46,); m/z(ES$^+$) found: [M + H]$^+$, 149.

**1** **2**

Preparation of iodomethyl 1-methylcyclopropanecarboxylate (2); sodium iodide (11.5 g, 77.0 mmol) was added to solution of chloromethyl 1-methylcyclopropanecarboxylate (3.8 g, 25.7 mmol) in acetone (25 ml) and stirred for 3 hr at 45°C. Reaction mixture was filtered, washed with acetone, and concentrated under vacuum. The residual oil was dissolved in diethyl ether (100 ml), washed with aqueous sodium bicarbonate and aqueous sodium thiosulfate, dried over $MgSO_4$, filtered, and concentrated to give crude iodomethyl 1-methylcyclopropanecarboxylate (4.6 g, 75%) $^1$H NMR, $\delta_H$ (CDCl$_3$, 400 MHz) 5.92 (2H, s), 1.31 (3H, s), 1.29 (2H, dd, J = 7.10), 0.74 (2H, dd, J = 6.88,) ; m/z(ES$^+$) found: [M + H]$^+$, 240.

**2** **3**

Preparation of (4-formyl-2-nitrophenoxy)methyl 1-methylcyclopropanecarboxylate (3); added solution of iodomethyl 1-methylcyclopropanecarboxylate (4.6 g, 19.2 mmol) in acetonitrile (50 ml) to silver oxide (13.7 g, 57.5 mmol) in acetonitrile (100 ml.) After stirring for 10 min at 0°C, solution of 4-hydroxy-3-nitro-benzaldehyde (4.0 g, 24.0 mmol) in acetonitrile (100 ml) was added dropwise and

stirred over night at 0°C. The reaction mixture was filtered through Celite then concentrated under vacuum to give crude (4-formyl-2-nitrophenoxy)methyl 1-methylcyclopropanecarboxylate (4.9 g, 91%) [1]H NMR, $\delta_H$ (CDCl$_3$, 400 MHz) 9.98 (1H, s), 8.35 (1H, d, J = 2.04), 8.10 (1H, dd, J = 8.68), 7.42 (1H, d, J = 8.68), 5.92 (2H, s), 1.29 (3H, s), 1.30 (2H, m), 0.79 (2H, t, J = 3.76,) ; m/z(ES[+]) found: [M + H][+], 280.

**3** → **4**

Preparation of (4-(hydroxymethyl)-2-nitrophenoxy)methyl 1-methylcyclopropanecarboxylate (4); NaBH$_4$ was added to a stirred solution of (4-formyl-2-nitrophenoxy)methyl 1-methylcyclopropane-carboxylate (4.9 g, 17.5 mmol) in chloroform (60 ml) and isopropanol (30 ml) at 0°C. Mixture was stirred at 0°C for an hour then at room temperature overnight. The reaction mixture was diluted with 150 ml of water and extracted with DCM (3 × 100 ml). The combined organic layer was washed with saturated brine (100 ml), dried over MgSO$_4$, filtered, and concentrated to residue oil. Purified by column chromatography (50% Ethyl acetate: hexane) to afford (4-(hydroxymethyl)-2-nitrophenoxy) methyl 1-methylcyclopropanecarboxylate (2.35 g, 48%) [1]H NMR, $\delta_H$ (CDCl$_3$, 400 MHz) 7.85 (1H, d, J = 2.12), 7.55 (1H, dd, J = 7.92), 7.42 (1H, d, J = 7.00), 5.82 (1H, s), 4.73 (2H, s), 1.31 (3H, s), 1.27 (2H, dd,J = 6.86), 0.76 (2H, dd, J = 6.9,); m/z(ES[+]) found: [M + H][+], 282.

**4** → **5**

Preparation of (2-nitro-4-((((4-nitrophenoxy)carbonyl)oxy)methyl)phenoxy)methyl 1-methylcyclopro-panecarboxylate (5); A solution of 4-nitrophenyl chloroformate (3.4 g, 16.7 mmol) in THF (10 ml) was added to a mixture of (4-(hydroxymethyl)-2-nitrophenoxy)methyl 1-methylcyclopropanecarboxylate (2.35 g, 8.35 mmol) and triethylamine (4.7 ml, 33.4 mmol) in THF (40 ml) and was stirred at 0°C. The reaction mixture was stirred in the dark at room temperature overnight. The reaction mixture was diluted with 150 ml of water and extracted with ethyl acetate (3 × 100 ml), the combined organic layer was washed with saturated brine (100 ml), dried over MgSO$_4$, filtered, and concentrated to residue oil. Purified by column chromatography (50% ethyl acetate: hexane) to afford (2-nitro-4-((((4-nitrophenoxy) carbonyl)oxy)methyl)phenoxy)methyl 1-methylcyclopropanecarboxylate (4.8 g, 65%) [1]H NMR, $\delta_H$ (CDCl$_3$, 400 MHz) 8.26 (2H, d, J = 9.24), 7.92 (1H, d, J = 2.20), 7.62 (1H, dd, J = 8.6), 7.36 (2H, d, J = 9.24), 7.28 (1H, d, J = 8.6) 5.82 (2H, s), 5.26 (2H, s), 1.29 (3H, s), 1.26 (2H, dd, J = 6.92), 0.75 (2H, dd, J = 6.74) ; m/z(ES[+]) found: [M + H][+], 447.

**5** → **6**

Preparation of CM-MK-801 (6); added and N,N-diisopropylethylamine (13 µl, 72 nmol) to mixture of (2-nitro-4-((((4-nitrophenoxy)carbonyl)oxy)methyl)phenoxy)methyl 1-methylcyclopropanecarboxylate (10 mg, 22 nmol) and (+)-MK 801 hydrogen maleate (8 mg, 24 nmol) in DMF (0.5 ml) at room temperature and stirred overnight. The reaction mixture was diluted with 100 ml of water and extracted with ethyl acetate (3 × 100 ml), combined organic layer was washed with saturated brine (100 ml), dried over $MgSO_4$, filtered, and concentrated. Purified by prep-HPLC (water: acetonitrile) to afford CM-MK-801 (4 mg, 32%) $^1H$ NMR, $\delta_H$ ($CDCl_3$, 400 MHz, temperature: 330K) 7.52 (1H, br s), 7.20 (1H, br d, J = 7.44), 7.15 (2H, m), 7.02 (3H, m), 6.94 (2H, m), 6.90 (2H, m), 6.73 (1H, m), 5.63 (2H, s), 5.30 (1H, d, J = 5.52), 4.96 (2H, dd, J = 33.8), 3.46 (1H, d, J = 16.86), 2.13 (3H, s), 1.17 (3H, s), 1.27 (2H, dd,J = 6.84), 0.59 (2H, dd, J = 6.7,); m/z($ES^+$) found: $[M + H]^+$, 529.

## Acknowledgements

This research was funded by the Howard Hughes Medical Institute. We thank C Magnus for assistance with electoporation; H Su for PLE plasmid construction. L Lavis provided technical advice from his earlier demonstration of the CM group for small molecule masking. YY performed the experiments. PL synthesized CM-MK801. YY and SMS designed the study. YY and SMS analyzed the data and wrote the paper.

## Additional information

### Funding

| Funder | Author |
| --- | --- |
| Howard Hughes Medical Institute (HHMI) | Scott M Sternson |

The funder had no role in study design, data collection and interpretation, or the decision to submit the work for publication.

### Author contributions

YY, Conception and design, Acquisition of data, Analysis and interpretation of data, Drafting or revising the article; PL, Analysis and interpretation of data, Drafting or revising the article, Contributed unpublished essential data or reagents; SMS, Conception and design, Drafting or revising the article

### Ethics

Animal experimentation: All experimental protocols were conducted according to United States National Institutes of Health guidelines for animal research and were approved by the Institutional Animal Care and Use Committee at Janelia Research Campus (protocol 13-92).

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
