## [Decision Letter]

Thank you for submitting your work entitled “Cell type-specific pharmacology of NMDA receptors using masked MK801” for peer review at *eLife*. Your submission has been favorably evaluated by a Senior Editor, a Reviewing editor, and three reviewers.

The following individuals responsible for the peer review of your submission have agreed to reveal their identity: Richard Palmiter (Reviewing editor) and Christian Lüscher (peer reviewer).

The reviewers have discussed the reviews with one another and the Reviewing editor has drafted this decision to help you prepare a revised submission.

This paper provides a new method for selective inactivation the NMDA receptors in specific cell types using a novel pharmacogenetic methodology.

All three reviewers believe that this is an excellent study that introduces a very useful new tool for studying the role of NMDA receptors in synaptic plasticity. However, there was concern that the paper goes overboard in trying to demonstrate a clear resolution to the role of NMDA receptors in cocaine-induced synaptic plasticity in dopamine neurons. They recommend that the paper emphasize the technique in the Introduction and Discussion, rather than the resolution of a problem.

For example, the conclusion related to dopamine neurons (Discussion) should be softened along the lines suggested below:

“These experiments with cell type-specific pharmacology provide direct evidence for the necessity of NMDA-Rs in dopamine neurons for the synaptic plasticity observed 3-5 hours after exposure to cocaine in vitro, leaving open the molecular explanation for the potentiation AMPA-Rs that occurs after many days of NMDA-R inactivity in dopamine neurons in vivo. […] Thus, it was not possible for us to determine if (or when) AMPA-R potentiation occurs after inactivation of NMDA-R in vivo using pharmacogenetic method described here.”

Reviewer #1:

This very interesting study by Yang et al. describes the use of enzymatic targeting of an NMDA-R antagonist (CM-MK801) achieved via neuronal cell-type-selectivity. The authors developed a new synthetic inert masked MK801 derivative that can be “unmasked” and activated inside the cells that have been previously successfully transfected with a virus that allowed the expression of the fundamental catalytic enzyme porcine liver esterase (PLE). Additionally, the authors tested for cellular specificity by using as a convenient experimental framework in utero electroporation of a plasmid that allowed co-expression of the PLE enzyme and of a fluorescent tag (mCherry). Subsequently, using brain slice electrophysiology from transfected mouse cerebral cortex the authors described convincing evidence of cellular specificity by showing a decreased in NMDA-R mediated excitatory postsynaptic current in a use-dependent manner in PLE^+^ cells and almost no effect in adjacent PLE^-^ cortical neurons. Moreover, the authors provided an additional experimental application of this new methodology. In fact, they used a brain slice model that is capable of recapitulating the cocaine mediated AMPA receptors potentiation in dopamine neurons, a phenomenon that is NMDA dependent. Elegantly, VTA brain slices from DATcre animals that have been infected with PLE and subsequently treated with CM-MK801 right before being incubated with cocaine are resistant to cocaine-induce plasticity. The study is thorough, very well done, and the data are both consistent and compelling.

Comments:

1) Citations should be provided for publications describing the generation of *Slc6a3*^*cre*^ animals.

2) Additionally, it would be nice to see a use-dependent blockade of the NMDA mediated currents elicited by the compound in the context of infected and non-infected dopamine cells residing into the VTA.

Reviewer #2:

This study by Yang, Lee, and Sternson investigates the viability of selective targeting of NMDA receptor inactivation by unmasking the antagonist MK-801. These investigators use a clever pharmacogenetic approach, combining cell-specific expression of the porcine liver esterase (PLE) with delivery of MK801 linked to 4-hydroxy-3-nitrobenzyl carbamate through a carboxymethylpropyl ester group. Using a cortical slice physiology preparation the authors demonstrate PLE/CM-MK801 blocks NMDAR-mediated synaptic currents. They also use this approach to show that NMDAR blockade in dopamine neurons prevents cocaine-evoked synaptic increases in AMPA receptors without causing the synaptic scaling previously shown to occur following genetic NMDAR inactivation. Overall the study is well written and logical. I believe this work is appropriate for publication in *eLife*, but would greatly benefit from the addition of a minor experiment to establish reversibility, or lack thereof.

1) The investigators use bath application for the delivery of CM-MK801 and perfuse over the slice throughout recordings. It is clear from Figure 2 that the effects take only a few minutes to reach maximal suppression of evoked NMDAR currents. It is well established that MK801 does not readily reverse in slice. An important experiment for the characterization of this approach is to establish the reversibility of the unmasked CM-MK801 compound.

Reviewer #3:

This is a demonstration of the power of a very cool new tool, a masked version of the MK801 NMDA antagonist that is unmasked enzymatically in cells that express PE under Cre control.

The study is certainly of interest to the researchers in the field and the data of very high quality. The selling point that this technique now allows for the identification of the locus of the NMDA receptor required for the induction of cocaine-evoked synaptic plasticity is probably overstated, but the conclusion very firm, at least for slice application of the drug.

Comments:

1) Most of the literature on cocaine-evoked synaptic plasticity used an ex vivo protocol whereby the drug was experiment injected and slices prepared the day after. This should be repeated here as well, also to demonstrate whether the masked MK801 works when injected directly into the brain (using the mCherry to visualise and quantify the proportion of neurons infected). This would also be a much better demonstration of the selectivity of the compound.

2) The description of [9], in the subsection “Cell type-specific NMDA-R pharmacology in cocaine-induced synaptic plasticity” is misleading. What the 2008 study by Engblom shows is that the ablation of GluN1 after completion of the postnatal development leads to a complete and selective loss of the NMDA currents in DA neurons. It also shows that when these animals were exposed to cocaine, no drug-evoked plasticity was observed (the IV curve of the AMPAR-EPSC remained linear and the amplitude of the spontaneous EPSCs remained unaffected. The conclusion of the paper thus was that the NMDA on DA neurons were required for the induction of the plasticity. This needs to be acknowledged in the Introduction. The present study is an important confirmation of the 2008 finding.

3) The study should be extended by looking at the current voltage relationship of the AMPAR-EPSC in DA neurons. Does the blockade of NMDARs also prevent rectification?

4) It is important to show that NMDA currents in non DA neurons of the VTA were unaffected.

[Editors' note: further revisions were requested prior to acceptance, as described below.]

Thank you for resubmitting your work entitled “Cell type-specific pharmacology of NMDA receptors using masked MK801” for peer review at *eLife*. Your submission has been favorably evaluated by a Senior Editor and a Reviewing Editor.

The revised version of your manuscript addresses some of the issues raised by the reviewers but fails to address their primary concern – namely that the tone of the Discussion fails to acknowledge that the conclusions regarding the role of NMDA receptors in promoting cocaine-mediated synaptic plasticity within dopamine neurons only applies to the conditions tested, i.e., the 3-5 hours after cocaine treatment of midbrain slices in vitro. If the authors want to make conclusions regarding the role of NMDA receptors in vivo, then the authors need to administer cocaine and the drug in vivo.

---

## [Author Response]

We thank the reviewers for their positive assessment of this study. We have made adjustments based on what was suggested by the Reviewing Editor and reviewers to emphasize the technique rather than the resolution of a problem. This includes:

a) In the subsection “Cell type-specific NMDA-R pharmacology in cocaine-induced synaptic plasticity” we now more clearly introduce the Zweifel and Engblom studies as well as the differences between them: both of which used genetic ablation of *Grin1* over 1 week but showed different results with respect to DA neuron synaptic potentiation, which was due to methodological challenges with AMP-R compensation. We then conclude our Results section with: “Therefore, using cell type-specific pharmacology, these experiments demonstrate that NMDA-Rs in dopamine neurons are necessary for cocaine-induced potentiation of synaptic AMPA-R currents in this cell type, *consistent with a prior report that examined this problem using Cre-ER* (9).”

b) In the Discussion with respect to our purpose for looking at NMDA-R in DA neurons, we changed the word *clarify* to *further examine*.

c) We added to the Discussion: “Because of this, it was not possible for us to determine if AMPA-R potentiation occurs after inactivation of NMDA-R in vivo using the chemogenetic method described here.”

d) The citation for the *Slc6a3*^*cre*^ mouse was added (25).

[Editors' note: further revisions were requested prior to acceptance, as described below.]

*The revised version of your manuscript addresses some of the issues raised by the reviewers but fails to address their primary concern – namely that the tone of the Discussion fails to acknowledge that the conclusions regarding the role of NMDA receptors in promoting cocaine-mediated synaptic plasticity within dopamine neurons only applies to the conditions tested, i.e., the 3-5 hours after cocaine treatment of midbrain slices in vitro. If the authors want to make conclusions regarding the role of NMDA receptors in vivo, then the authors need to administer cocaine and the drug in vivo*.

We agree with the reviewers that the focus should be on the cell type-specific pharmacology method. The choice to examine NMDA-R-mediated synaptic potentiation after cocaine administration was based on a desire to examine the utility of the system in a biological system. A nice example of the challenge with ion channel manipulations with genetic methods was the considerable difficulty that was reported by the Palmiter and Lüscher groups in 2008 for DA neuron *Grin1* ablation. Lüscher ultimately managed to avoid compensatory increases in AMPA-R rectification using Cre-ER in vivo. We thought that confirmation of this in vivo result, even with an ex vivo application of our tools would be a nice demonstration of the strengths of our method, and it also led to us discover an in vivo limitation of the current implementation of the method due to limited aqueous solubility. In our last revision, we made a number of changes to reflect this point. Now, we have made further modifications to emphasize the ex vivo aspects of our study and the confirmatory nature of it, relative to the Engblom study.

Additional changes include the final sentence of the Results, which now reads:

“Therefore, using cell type-specific pharmacology *in an ex vivo preparation*, these experiments demonstrate that NMDA-Rs in dopamine neurons are necessary for cocaine-induced potentiation of synaptic AMPA-R currents in this cell type ex vivo, which is consistent with a prior report that examined this problem using Cre-ER in vivo (9).”

In the first paragraph of the subsection “Cell type-specific NMDA-R pharmacology in cocaine-induced synaptic plasticity”, the sentence has been changed to remove the word “concretely”:

“However, it has been *challenging to establish* the functional importance of NMDA-Rs specifically in dopamine neurons for cocaine-induced synaptic AMPA-R potentiation, as opposed to a possible indirect process involving NMDA-Rs on other cell types.”

The word “concretely” was removed, and the sentence simply reflects the considerable difficulty that was reported in the Lüscher and Palmiter papers to avoid compensatory effects for *Grin1* deletion.

In the Discussion, we deleted a clause in the first sentence and we refer to our work as additional evidence for the necessity of NMDA-Rs for synaptic potentiation:

“We applied this to dopamine neurons to further examine the role for NMDA-Rs in cocaine-induced synaptic potentiation, which has been investigated in several pharmacological (20; 1) and genetic studies (9; 26). These experiments with cell type-specific pharmacology provide *additional* evidence for the necessity of NMDA-Rs in cocaine-induced long-term synaptic potentiation in dopamine neurons ex vivo.”

One sentence later we write:

“Because of this, it was not possible for us to determine if AMPA-R potentiation occurs after inactivation of NMDA-R in vivo using the chemogenetic method described here. Therefore, additional improvements to cell type-specific pharmacological techniques are needed to facilitate manipulations in mammalian brains in vivo.”

Moreover, the next paragraph is all about the utility of this tool as an ex vivo method for cell type-specific NMDA-R blockade. Based on these changes it seems that it would be difficult for the reader to now confuse our conclusions as referring to anything but an ex vivo finding and also as a confirmation of a prior in vivo finding by Engblom et al.
